# A Study on Impact Force Detection Method Based on Piezoelectric Sensing

**DOI:** 10.3390/s22145167

**Published:** 2022-07-10

**Authors:** Jianli Liu, Chuang Hei, Mingzhang Luo, Dong Yang, Changhe Sun, Ankang Feng

**Affiliations:** 1School of Electronics and Information, Yangtze University, Jingzhou 434023, China; 201972315@yangtzeu.edu.cn (J.L.); lmz@yangtzeu.edu.cn (M.L.); chsun@yangtzeu.edu.cn (C.S.); 201606279@yangtzeu.edu.cn (A.F.); 2School of Mechanical Engineering, Yangtze University, Jingzhou 434023, China; dong.yang@yangtzeu.edu.cn; 3National Demonstration Center for Experimental Electrotechnics and Electronics Education, Yangtze University, Jingzhou 434023, China

**Keywords:** impact force, impact location, numerical simulation, piezoelectric sensor, impact detection

## Abstract

Impact force refers to a transient phenomenon with a very short-acting time, but a large impulse. Therefore, the detection of impact vibration is critical for the reliability, stability, and overall life of mechanical parts. Accordingly, this paper proposes a method to indirectly characterize the impact force by using an impact stress wave. The LS-DYNA software is utilized to establish the model of a ball impacting the steel plate, and the impact force of the ball and the impact response of the detection point are obtained through explicit dynamic finite element analysis. In addition, on this basis, a correspondence between the impact force and the impact response is established, and finally, an experimental platform for impact force detection is built for experimental testing. The results obtained by the finite element method are in good agreement with the experimental measurement results, and it can be inferred that the detected piezoelectric signal can be used to characterize the impact force. The method proposed herein can guide the impact resistance design and safety assessment of structures in actual engineering applications.

## 1. Introduction

External impact events on structures such as aircraft and wind turbine blades are a common phenomenon in engineering practice, such as bird strikes, hailstorms, falling object impacts, and other similar impact events. Shock vibrations generated by impacts can have a significant impact on equipment or mechanical components, mainly in terms of reduced reliability, stability, and service life, or even serious damage to the structure. Therefore, it is very necessary to know the exact value of the load when designing the structure. By measuring the impact force on machinery and equipment, the force conditions and working conditions can be analyzed, the rationality of the structural design scheme can be verified, the cost of the structure can be effectively controlled, and damage to the structure can be reduced. By measuring the impact force in production, the safety condition of the equipment in the production process can be understood timely and accurately to ensure its normal operation. Since the thorough inspection of large structures can be time- and resource-consuming, this problem would be greatly alleviated if automatic detection and identification were possible. Therefore, the rapid and accurate detection of impact forces and the localization of impact locations are extremely important topics of research significance.

Impact force recognition mainly includes impact position estimation and force time history reconstruction. For impact location recognition, Liang et al. [1] introduced a distributed coordination algorithm that combines the benefits of triangulation and inverse analysis, and these two methods work together to increase the accuracy and real-time performance of impact location. In another work, Chang et al. [2,3,4] proposed a method to estimate the impact position by calculating the centroid of power distribution on the entire structure using a sensor signal. Giurgiutiu [5] considered shock and acoustic emissions in structural health monitoring. Zhu [6] proposed a rapid positioning method for underwater pipeline impacts based on the arrival time and group velocity of ultrasonic guided waves. Similarly, Jang et al. proposed a triangulation method based on acoustic emission to locate the impact source [7]. Progressively, Alajlouni et al. [8] proposed a multi-point positioning method to address the dispersive guided wave, which is used to locate the impact force in a dispersed floor. Chen et al. [9] proposed a novel arrival time calculation method that does not require the manual selection of thresholds to determine the location of the two-dimensional impact on the pipeline. Although this method is simple and does not require any complicated algorithms, the accuracy of the results depends largely on the number of sensors. In addition, with the development of artificial intelligence, artificial neural networks have also been used to detect the impact locations. Worden et al. used an artificial neural network to locate the impact position successfully [10,11,12]; however, they failed to detect the magnitude of impact force.

On the other hand, for impact force recognition, M. Thiene et al. [13] used surface-bonded piezoelectric sensors to conduct impact tests and evaluated the transfer function to achieve impact force reconstruction. In addition, Sun et al. [14] used an improved back-analysis technique based on the finite element method and modal superposition method to identify the impact force. Tian et al. [15] utilized piezoelectric ceramic sensors to detect the dynamic response of pipes under impact loads. In addition to these, researchers have also developed other kinds of effective methods, i.e., analytical solutions [16,17], finite element methods [18,19], conjugate gradient methods [20,21], basis function expansion methods [22,23], regularization methods [24,25,26], experimental detection methods [4,5,27,28], and artificial neural networks [29] have all been developed for impact recognition tasks.

In recent years, piezoelectric sensors have been widely used in structural health detection, stress monitoring, and other relevant fields. Luo et al. [30] and Xu et al. [31] used the arrival time method of piezoelectric sensors to evaluate the concrete filling status of carbon-fiber composite materials. Likewise, Xu et al. [32] proposed a new method to detect the pipeline defect location, based on the time-reversal method and multi-scale denoising. Relying on piezoelectric sensors, Li et al. [33] proposed a method to quantitatively evaluate the debonding of concrete-filled steel tube members. Gong et al. [34] proposed an automatic extraction algorithm for the stress wave reflection period based on image processing. In another work, Hei et al. [35] presented a method to quantitatively evaluate the bolt connection state by using a single piezoelectric ceramic transducer and a wake wave. Although the above research results are practical and effective, a large number of experiments are required in the research process, which may result in a waste of resources.

In this paper, a method to characterize impact force with a stress wave is proposed, a finite element model of an impact force detection system is designed, and the functional relationship between impact force and stress wave is obtained. The creation of the impact force model plays a guiding role in the design of the impact resistance of the structure and the safety evaluation of the mechanical equipment in the engineering practice. At the same time, a shock signal extraction method is proposed and packaged which can be called by other programs to realize the identification of shock position and shock load. This method can be used for the structural health monitoring of shock forces.

The rest of this article is organized as follows: Section 2 reports the theoretical research and simulations. Section 3 presents the development of an experimental platform to conduct experiments and obtain experimental results. Section 4 discusses the results of the numerical simulation and experiment. Finally, the work is concluded in Section 5.

## 2. Materials and Methods

### 2.1. Positioning Principle

In any impact event, surface waves are generated. When a surface wave reaches the sensor, the local strain signal is recorded as a time-varying strain or voltage. These local signals contain a set of features corresponding to the impact location and force, and therefore, they can be used to characterize the impact events [36]. In this paper, the impact of a steel ball on a steel plate is studied as an example. A two-dimensional coordinate system of the x–y axis is defined on the plane, as shown in Figure 1. Suppose the position of the sensor A is (x_1_, y_1_), the position of the sensor B is (x_2_, y_2_), the position of sensor C is (x_3_, y_3_), and the position of sensor D is (x_4_, y_4_). The distances from the signal source *p* to the sensor are s1, s2, s3, and s4, respectively. The arrival times of the signal source p to the sensor are T_1_, T_2_, T_3_, and T_4_, respectively. The stress wave propagating along the steel plate generated by the steel ball impacting the steel plate can be detected by the sensor installed on the steel plate. The arrival time of the impact signal detected by the sensor is extracted [37], and the location of the impact point can be calculated according to the arrival time difference method.

According to the impact positioning principle in Figure 1, Equation (1) can be obtained.
(1)$$\{\begin{array}{c} \left(T_{1}-T_{2}\right)\times C=s1-s2\\ \left(T_{1}-T_{3}\right)\times C=s1-s3\\ \left(T_{1}-T_{4}\right)\times C=s1-s4 \end{array}$$
where C denotes the speed of wave propagation. Knowing the coordinates of the impact source *p* (x, y) and the coordinates of the sensors S_A_ (x_1_, y_1_), S_B_ (x_2_, y_2_), S_C_ (x_3_, y_3_), S_D_ (x_4_, y_4_), there are:(2)$$\{\begin{array}{c} s1=\sqrt{{\left(x-x_{1}\right)}^{2}+{\left(y-y_{1}\right)}^{2}}\\ s2=\sqrt{{\left(x-x_{2}\right)}^{2}+{\left(y-y_{2}\right)}^{2}}\\ s3=\sqrt{{\left(x-x_{3}\right)}^{2}+{\left(y-y_{3}\right)}^{2}}\\ s4=\sqrt{{\left(x-x_{4}\right)}^{2}+{\left(y-y_{4}\right)}^{2}} \end{array}$$

The system of Equation (3) can be obtained from the system of Equation (1) and the system of Equation (2).
(3)$$\{\begin{array}{c} \left(T_{1}-T_{2}\right)\times C=\sqrt{{\left(x-x_{1}\right)}^{2}+{\left(y-y_{1}\right)}^{2}}-\sqrt{{\left(x-x_{2}\right)}^{2}+{\left(y-y_{2}\right)}^{2}}\\ \left(T_{1}-T_{3}\right)\times C=\sqrt{{\left(x-x_{1}\right)}^{2}+{\left(y-y_{1}\right)}^{2}}-\sqrt{{\left(x-x_{3}\right)}^{2}+{\left(y-y_{3}\right)}^{2}}\\ \left(T_{1}-T_{4}\right)\times C=\sqrt{{\left(x-x_{1}\right)}^{2}+{\left(y-y_{1}\right)}^{2}}-\sqrt{{\left(x-x_{4}\right)}^{2}+{\left(y-y_{4}\right)}^{2}} \end{array}$$

Combined with the propagation speed of the stress wave in the thin plate, the time of the stress wave signal reaching the sensor can be automatically extracted by MATLAB programming, and finally, the impact position can be calculated by Equation (3).

The voltage signal detected by the sensor can reflect the change of the shock energy at the moment of shock, that is when the signal voltage is the largest, that is, the moment when the shock energy is the strongest. Studies have shown that the shock process can usually be represented as a half-sine pulse shape. In this paper, the impact process of the steel ball impacting the steel plate is roughly regarded as a 1/2 cycle sinusoidal pulse signal as shown in Figure 2. If the height of the steel ball is h and the mass is m, according to the functional principle, the speed of the steel ball before it touches the steel plate is $\sqrt{2\mathrm{gh}}$, where g = 9.8 m/s. After contacting the steel plate, the velocity of the steel ball decreases so rapidly that it is negligible. Therefore, the impulse I of the impact process can be expressed as:I = *F_I_* × t = m × v(4)
where $F_{I}\times t$ represents the integral of the impulse $f_{It‘}$ in the T time period $\int_{0}^{t‘}f_{It‘}$t‘dt‘, m is the mass of the impact source, and v is the impact velocity. Since v = $\sqrt{2\mathrm{gh}}$, the calculation formula of the theoretical value of the impact force can be obtained:(5)$$F=\frac{m\sqrt{2\mathrm{gh}}}{t}$$

### 2.2. Numerical Simulation

Impact force detection simulation process;

In the numerical simulation, LS-DYNA was first used for preprocessing, and 3D cell types were created for finite element analysis according to the physical material properties and dimensions. The impact model of the impact platform is shown in Figure 3, and the impact platform composition and material parameters are shown in Table 1. After meshing, the finite element model is generated as shown in Figure 4, and the contact mode, constraints, and impact velocity of the impact model are set; then, the LS-DYNA solver is used to solve the calculation and obtain the finite element analysis results of the impact model. Finally, LS-PRE POST software is used to view the finite element analysis results and evaluate the analysis.

LS-PRE POST is an advanced finite element pre- and post-processing software developed specifically for LS-DYNA. After the explicit kinetic analysis calculation, the process of the steel ball impacting the steel plate will be recorded and saved as a corresponding file, and we can reproduce the impact process by viewing the corresponding impact force file with LS-PRE POST software.

LS-DYNA simulation steps;

In the impact force detection system, in order to avoid the occurrence of possible non-unique solutions, the impact position is usually determined first, and then the impact load is identified. In order to observe the impact response of different impact positions on the impact platform, different impact points are selected on the impact platform and marked as position 1, position 2, position 3; their coordinates are P1 (0, 0), P2 (30, 30), and P3 (50, 50). The impact position and the position distribution of detection points A, B, C, and D are shown in Figure 5.

Step 1: Select impact position 1, apply 1 m/s, 1.5 m/s, 2 m/s, 2.5 m/s, and 3 m/s on the steel ball to impact the steel plate, obtain the relevant data of stress and impact force at the detection point, and establish the correspondence between impact force and stress wave.

Step 2: Move the impact point to position 2 and position 3 in turn, and set the steel ball to impact the steel plate at 1 m/s, 1.5 m/s, 2 m/s, 2.5 m/s, and 3 m/s, respectively. Again, in our experiment, the relevant data was recorded and data analysis and processing were performed.

LS-DYNA simulation results;

According to wave propagation theory, the impact signal begins to travel uniformly along all directions of the plate from the impact source on the same-sex steel plate. When a steel ball strikes a steel plate at varying speeds, the stress waves produced by the impact differ. Figure 6 depicts the stress nephogram under different impact speeds.

Under different impact speeds, the steel ball exhibits different impact forces, and the stress waves detected by the detection points are also different. The LS-PREPOST post-processing software was used for analysis and processing, and the impact histories at different speeds of 1 m/s, 1.5 m/s, 2 m/s, 2.5 m/s, and 3 m/s were obtained, as shown in Figure 7. The stress wave at the detection point is shown in Figure 8.

Then. the impact point was adjusted to position 2 and position 3 in turn, and the steel ball was set to impact the steel plate with a speed of 1 m/s, 1.5 m/s, 2 m/s, 2.5 m/s, and 3 m/s, respectively. The corresponding data were recorded and analyzed to obtain the stress distribution at position 2 as shown in Figure 9, the impact force curve as shown in Figure 10, and the stress curve at the detection point as shown in Figure 11, as well as the stress distribution at position 3 as shown in Figure 12, the impact force curve as shown in Figure 13, and the stress curve at the detection point as shown in Figure 14.

It can be seen from Figure 6, Figure 9 and Figure 12 that the stress distribution on the steel plate differs under different velocity impacts; the faster the impact velocity, the larger the stress response on the steel plate at the same moment. This feature can also be derived from the impact force curves and stress curves when impacted at position 1, position 2, and position 3. Based on this, the correspondence between the impact force and stress wave can be established based on the stress curves of the detection points at impact position 1, position 2, and position 3 and the impact force curves, which are expressed as FEM1, FEM2, and FEM3, as shown in Figure 15; the corresponding functional relationships between the fitted impact force and stress wave are shown in Table 2. This provides a theoretical basis for the experimental detection of the impact force.

It can be seen from Figure 15 that there is a linear relationship between the impact force and the stress wave. When the impact position is different, the stress of the test point is different, but the slope of the linear relationship between the impact force and the stress wave is the same (as shown in Table 2); under the same impact force, the closer to the detection point, the more obvious the stress response.

## 3. Experimental Research and Results

### 3.1. Experimental Setup

An experiment was conducted to study the relationship between impact force and impact response when a small ball was struck against a steel plate. The impact point setting is consistent with Figure 5, and the four sensors A, B, C, and D were installed symmetrically at equal distances, consistent with the four detection positions mentioned in Figure 5, and are used to detect the impact signals. The setup for the impact detection device is shown in Figure 16, which includes a steel ball with a diameter of 40 mm and a weight of 260 g; a steel plate with a length of 275 mm, a width of 275 mm, and a height of 3 mm; a scale ruler with a length of 1 m; a multi-channel data acquisition card with a maximum sampling rate of 500 ksps; four quartz crystal piezoelectric sensors with a diameter of 14 mm and a thickness of 0.3 mm; and one computer.

### 3.2. Experimental Process

In the finite element analysis, the steel ball impacting the steel plate can set the velocity directly, but the impact velocity should be converted to height to represent the experimental test. The material of the steel ball and the steel ball used in the finite element simulation is the same, and this can be verified by the virtual prototype ADAMS. It is also possible to use ADAMS to calculate the heights corresponding to the velocity impacts of 1 m/s, 1.5 m/s, 2 m/s, 2.5 m/s, and 3 m/s as 0.05 m, 0.12 m, 0.20 m, 0.32 m, and 0.45 m, respectively (for the convenience of presentation, the velocity description is still used in the subsequent experiments).

The experimental procedure for detecting the impact force is as follows:

Step 1: When the experimental device is connected and working normally, use small balls to impact the steel plate at position 1 with speeds of 1 m/s, 1.5 m/s, 2 m/s, 2.5 m/s, and 3 m/s, respectively. Next, obtain the voltage signals of four sensors, A, B, C, and D, under different impact speeds. For the arrival time of the extracted signal, calculate the time difference between the impact point and each sensor to verify the impact position. Subsequently, calculate the impact force under different impact speed states, and obtain the relationship between the impact force and the voltage signal.

Step 2: Adjust the impact position to position 2, position 3, position 4, position 5, and position 6, as shown in Figure 17. Repeat step 1 to obtain multiple sets of experimental data to verify the correctness of the detection method.

### 3.3. Experimental Results

In the experiment, the steel ball was set to impact the steel plate at 1 m/s, 1.5 m/s, 2 m/s, 2.5 m/s, and 3 m/s at position 1, and the sensor captured the voltage signals under different speeds. The voltage waveform for sampling 20,000 valid data points at a sampling rate of 200 ksps is shown in Figure 18. As shown in Figure 19, the impulse under each impact velocity is calculated. It can be seen from Figure 18 and Figure 19 that the impulse under different impact speeds changes linearly, and the detected response voltage increases with the acceleration of impact speed.

Similarly, the impact was adjusted to position 2, position 3, position 4, position 5, and position 6 in turn, and the impact experiments were conducted at 1 m/s, 1.5 m/s, 2 m/s, 2.5 m/s, and 3 m/s, respectively, to obtain the corresponding voltage signals. The relationship between the impact force and the voltage signal was established according to the voltage signals detected at impact position 1, position 2, and position 3, which are expressed as Test1, Test2, and Test3 in Figure 20, and the functional relationship between the fitted impact force and voltage is shown in Table 3. As shown in Table 4, the actual impact position of the small ball impacting the steel plate is compared with the detected impact position. As can be seen from the table, the experimental results do not differ much from the real coordinates, and the maximum axial relative error does not exceed 11%. The relative error of impact position estimation is shown in Figure 21.

## 4. Discussion

The experiments in this paper are non-destructive. In the numerical simulation and experimental test, the steel ball impacts the steel plate at a low speed; that is, the relationship between the stress and the strain of the steel plate are in a linear stage. At this time, the strain is proportional to the load, and the numerical simulation and test results are consistent with the predicted results. The impact force increases with the acceleration of the impact velocity, and the two are linear. After exceeding the linear stage, the stress will no longer change linearly with the strain. For a short-range beyond the linear stage, the steel plate is still elastic, which means that the deformation of the steel plate can be restored after the load is removed. When the steel plate exceeds the elastic limit, plastic deformation occurs, which means that the deformation cannot be restored by removing the load. The variation law of elastic and plastic stages will be studied in subsequent work.

In the numerical simulation, a finite element model for impact force detection was created. The impact force and impact response were obtained by explicit kinetic finite element analysis, and a functional relationship between the impact force and impact response was established. The accuracy of the results depends on the number of finite cells, i.e., the optimal solution of the model meshes. The creation of the finite element model for impact force detection can effectively reduce the number of experimental prototypes, and it can help engineering designers determine in advance the structural practical conditions of the product under various operating conditions, to understand in time the response of the structure under different types of dynamic loads. Numerical calculation by FEM can determine the impact of the linear and elastic limits of the structure under different materials, thicknesses, and loads, which can provide a reference for engineering design and facilitate the evaluation and parameter control of the structure.

In the experiments, steel balls are used to impact steel plates to verify the conclusions of the numerical simulations. The accuracy of the experimental results depends on the accuracy of the arrival time of the signal received by the sensor. A quartz sensor with high accuracy is selected to capture the impact signal; after data processing, the arrival time of the effective signal is obtained; the impact location principle is used to calculate the impact position; the impact force is identified by the detected voltage signal, and thus the functional relationship between the impact force and the voltage signal is obtained. The method can be used for structural health monitoring and nondestructive assessment.

The impact positioning and the accuracy of the impact force are influenced by several factors, including impact mass, impact velocity, impact time, impact area, etc. Changing the value of each quantity can change the magnitude of the impact force. In numerical simulations and experimental tests, a steel ball with a mass of 260 g and a diameter of 40 mm was used to impact a steel plate with a volume of 275 × 275 × 2 mm. The contact between the steel ball and the steel plate was point contact, and the factors affecting the magnitude of the impact force were relatively single. Changing the impact surface between the two objects or changing the mass of the steel ball, the size and thickness of the steel plate present a law of change that needs further study.

It can be seen from Figure 8 and Figure 13 that the results obtained by the numerical simulation are very similar to the experimental measurement results, and both have a linear relationship. This conclusion can also be seen from the functional relationship (Table 2) obtained by numerical simulation and the functional relationship (Table 3) obtained by experiment. The difference is that the numerical simulation uses stress to characterize the impact force, while the experimental result uses voltage to characterize the impact force (the comparison of impact force between numerical simulation and the experimental test is shown in Table 5). At the same time, the slopes of the fitting curves of the numerical simulation and experimental test results are the same, and the closer the impact position is to the detection point, the smaller the intercept of the functional relationship between the impact force and the impact response. This shows that it is feasible to use the sensor to detect the stress wave to characterize the impact force, but there are still some errors, which may be due to the following factors:(1)There may be differences between the real parameters of the steel ball and the steel plate and the parameters used in the finite element analysis. When performing explicit dynamic finite element analysis, it may be affected by damping and inertial forces;(2)In the experiment, the extraction of the arrival time of the signal and the dispersion characteristics of the stress wave affect the calculation results;(3)The degree of coupling between the steel ball and the steel plate in the experiment will affect the magnitude of the impact force.

## 5. Conclusions

This paper proposes a method for the indirect characterization of the impact force by stress waves and verifies the feasibility of the method through finite element analysis and experiments. The finite element analysis of the impact detection model shows that the faster the impact velocity, the higher the impact force under the same conditions; the higher the impact force applied, the greater the response of the corresponding detection point; and the smaller the distance from the impact point to the detection point, the greater the impact response of the detection point. It can effectively reduce the number of experimental prototypes and save the experimental cost and predict the linear and elastic limits of the structure, but the change law of the elastic and plastic phases of the structure needs further study.

In the experiments, the transient behavior of a steel ball impacting a steel plate is used to reconstruct the impact history, and the voltage signal of the steel ball impact is obtained using a piezoelectric transducer to establish the functional relationship between impact force and voltage. The experimental results are in good agreement with the numerical simulation results, indicating the feasibility and effectiveness of the indirect characterization of the impact force method using the piezoelectric sensor to detect the stress wave signal. The method can be used for structural health monitoring and nondestructive assessment. However, other factors affecting the magnitude of impact force need to be further investigated.

## Figures and Tables

**Figure 1 sensors-22-05167-f001:**
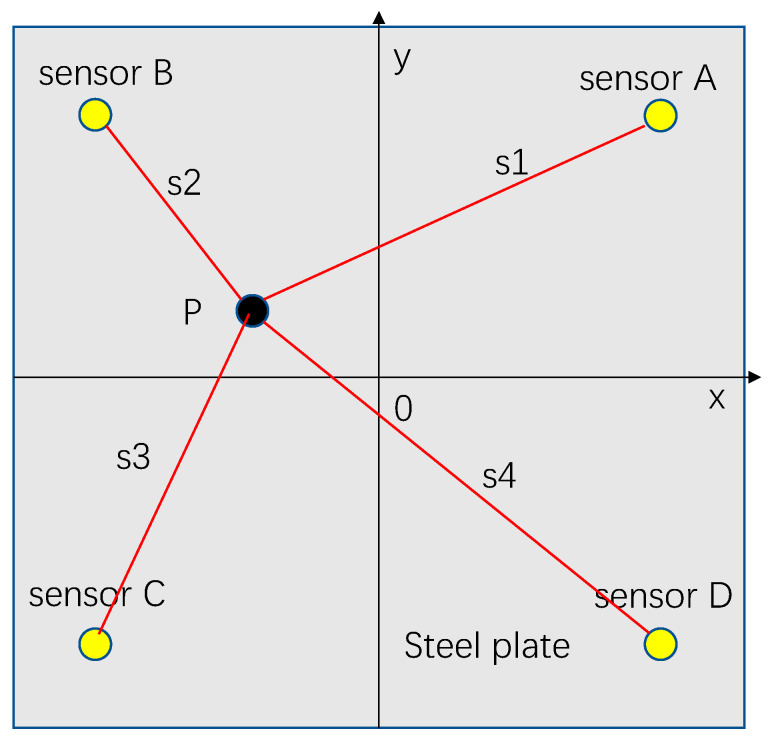
Position distribution of sensors.

**Figure 2 sensors-22-05167-f002:**
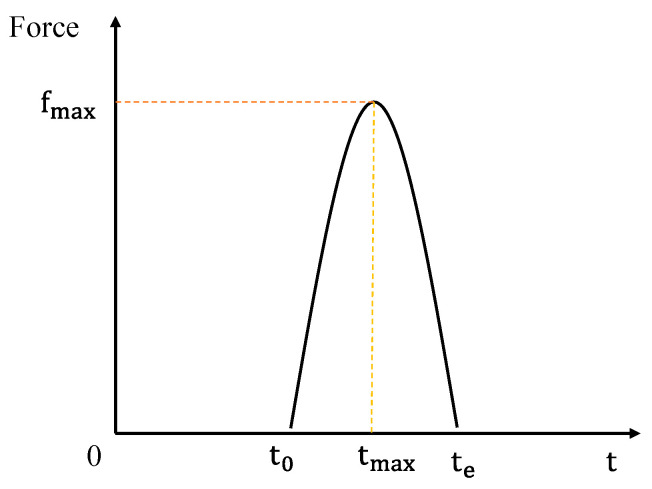
Shock history of positive half cycle.

**Figure 3 sensors-22-05167-f003:**
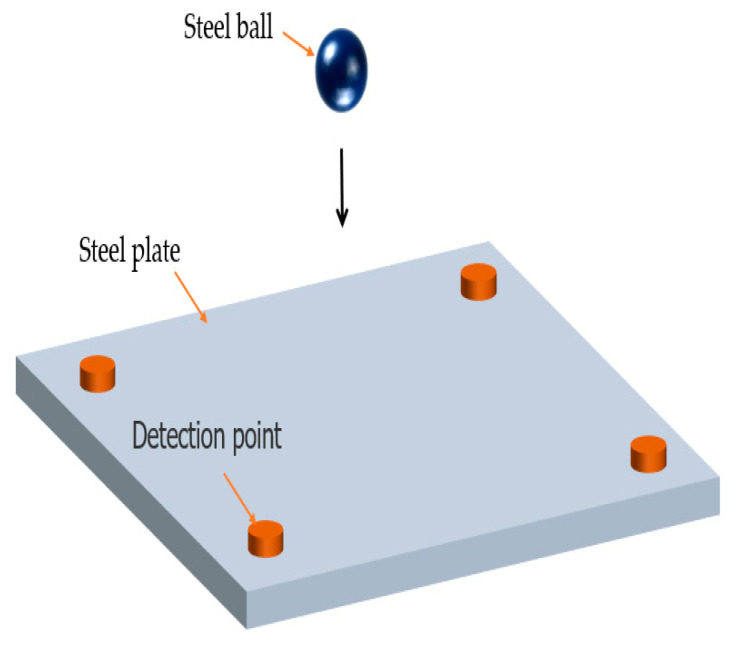
Impact model diagram.

**Figure 4 sensors-22-05167-f004:**
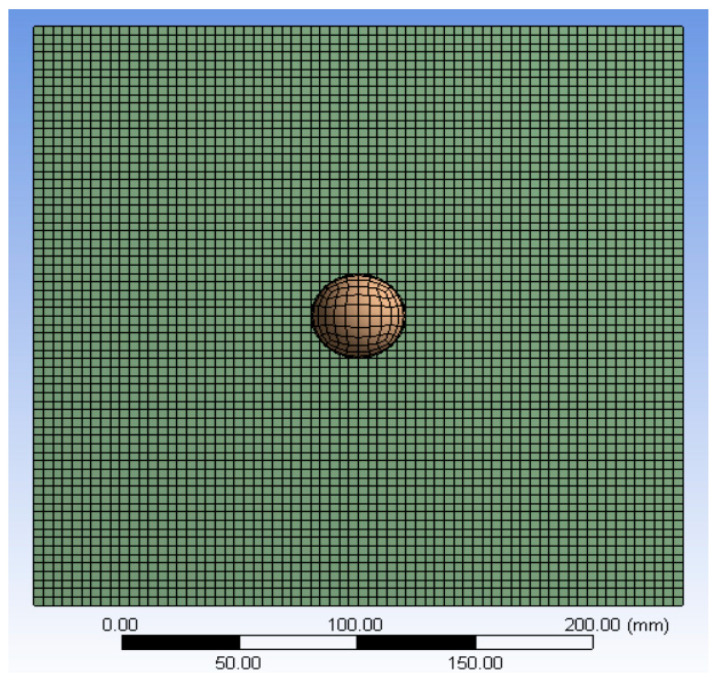
Meshing.

**Figure 5 sensors-22-05167-f005:**
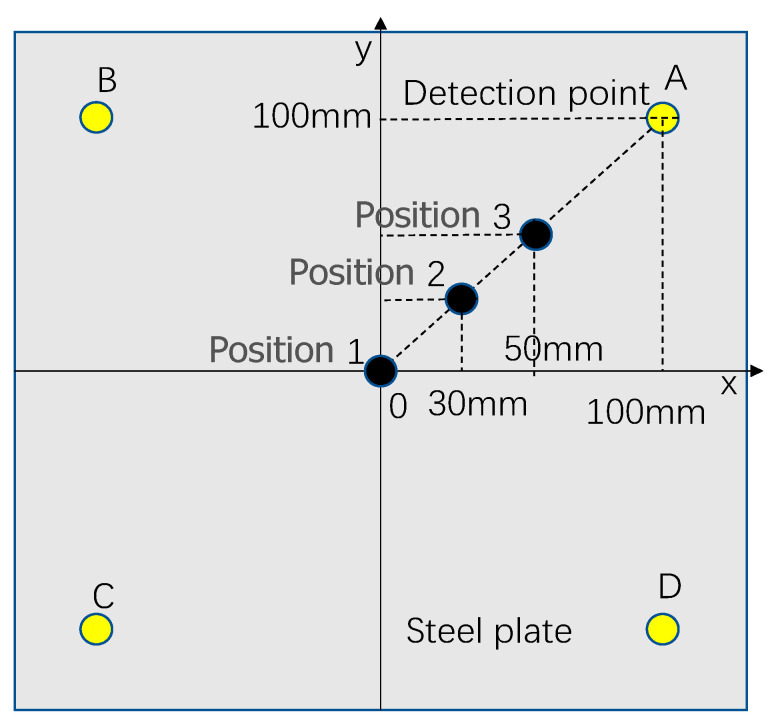
Distribution of impact positions and detection positions.

**Figure 6 sensors-22-05167-f006:**
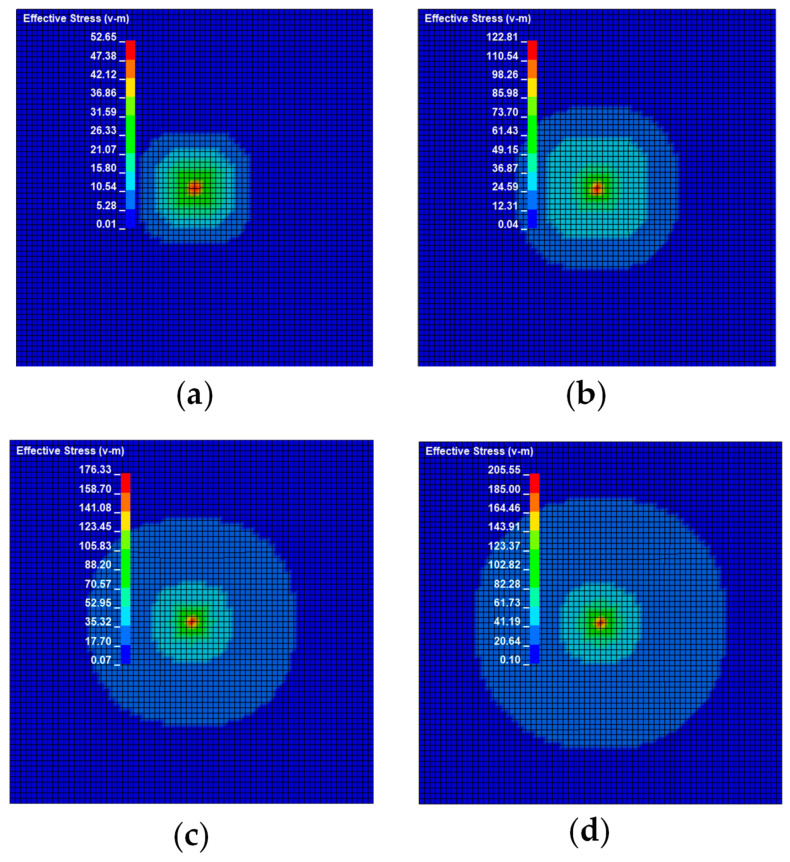
Stress cloud diagram of impact response at different velocities at position 1: (**a**) shock response stress cloud diagram at an impact speed of 1 m/s; (**b**) shock response stress cloud diagram at an impact speed of 1.5 m/s; (**c**) shock response stress cloud diagram at an impact speed of 2 m/s; (**d**) shock response stress cloud diagram at an impact speed of 2.5 m/s.

**Figure 7 sensors-22-05167-f007:**
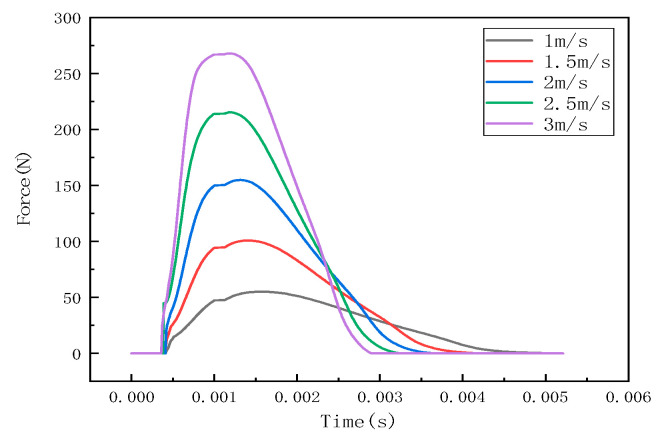
Impact force curve at position 1.

**Figure 8 sensors-22-05167-f008:**
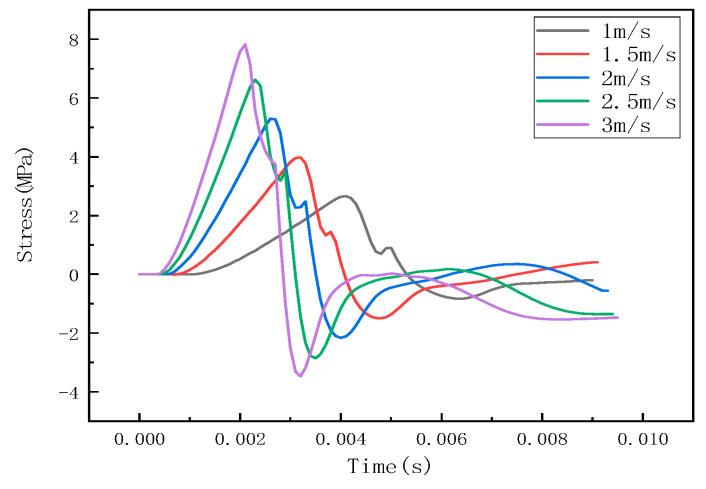
Stress curve of detection point at impact at position 1.

**Figure 9 sensors-22-05167-f009:**
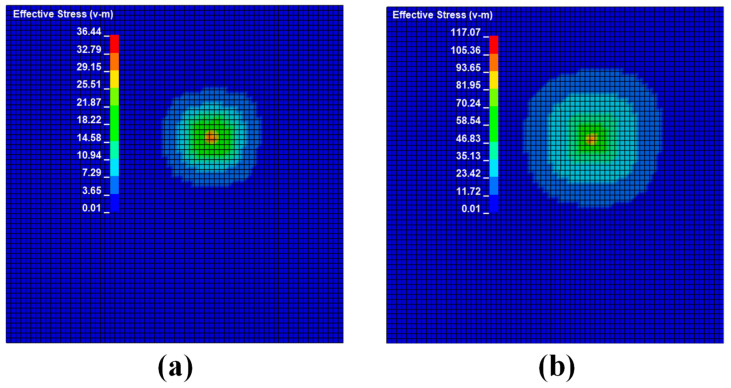
Stress cloud diagram of impact response at different velocities at position 2: (**a**) shock response stress cloud diagram at an impact speed of 1 m/s; (**b**) shock response stress cloud diagram at an impact speed of 1.5 m/s; (**c**) shock response stress cloud diagram at an impact speed of 2 m/s; (**d**) shock response stress cloud diagram at an impact speed of 2.5 m/s.

**Figure 10 sensors-22-05167-f010:**
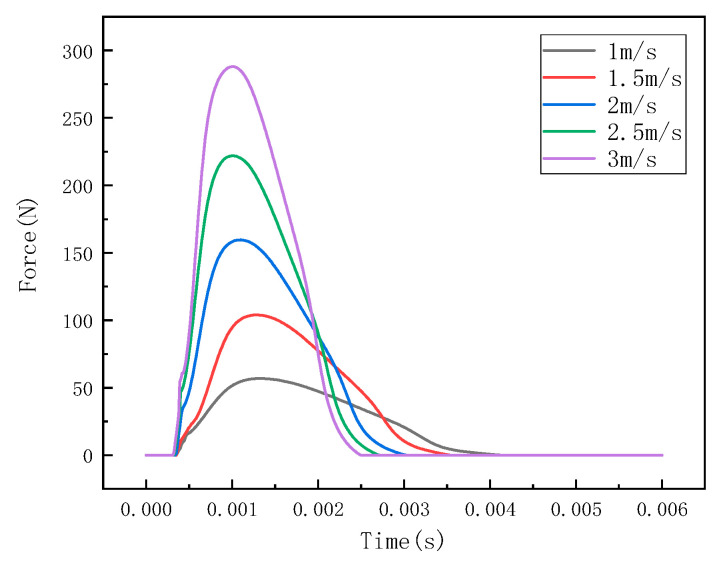
Impact force curve at position 2.

**Figure 11 sensors-22-05167-f011:**
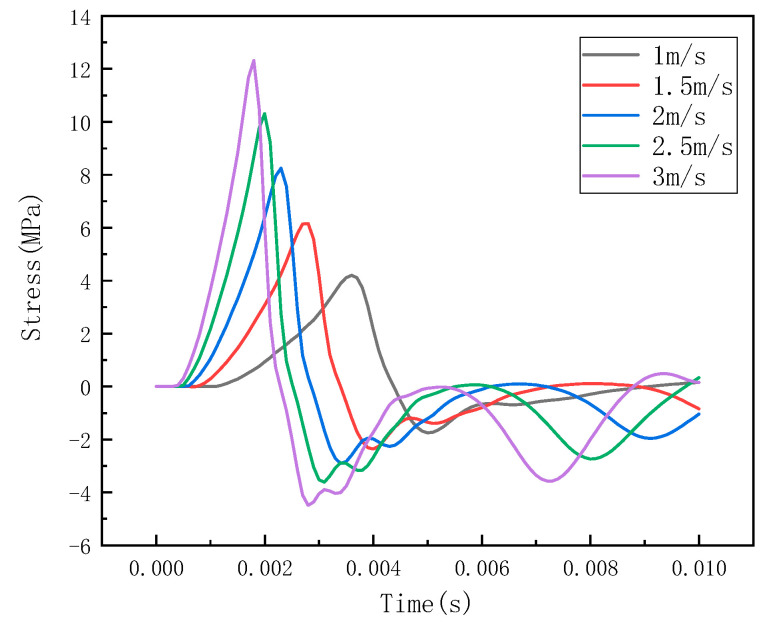
Stress curve of detection point at impact at position 2.

**Figure 12 sensors-22-05167-f012:**
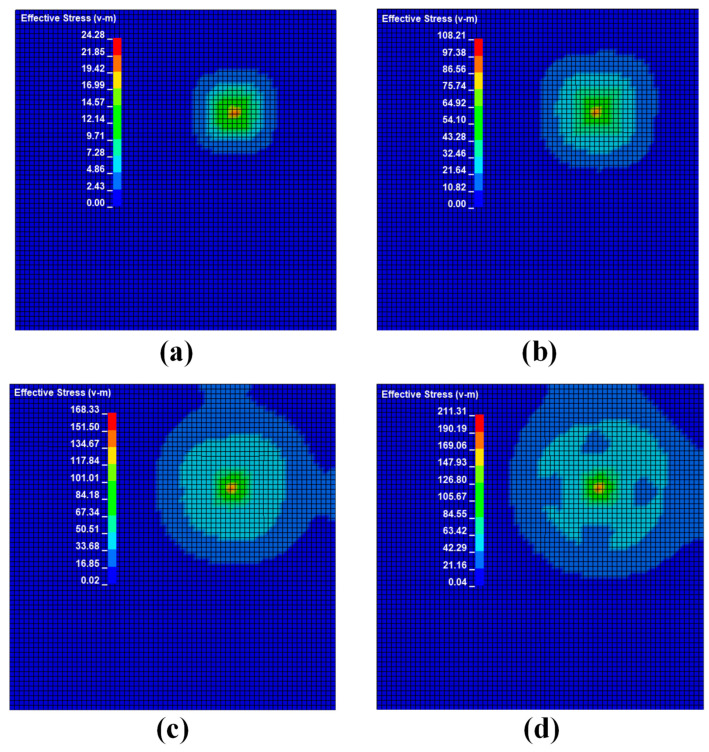
Stress cloud diagram of impact response at different velocities at position 3: (**a**) shock response stress cloud diagram at an impact speed of 1 m/s; (**b**) shock response stress cloud diagram at an impact speed of 1.5 m/s; (**c**) shock response stress cloud diagram at an impact speed of 2 m/s; (**d**) shock response stress cloud diagram at an impact speed of 2.5 m/s.

**Figure 13 sensors-22-05167-f013:**
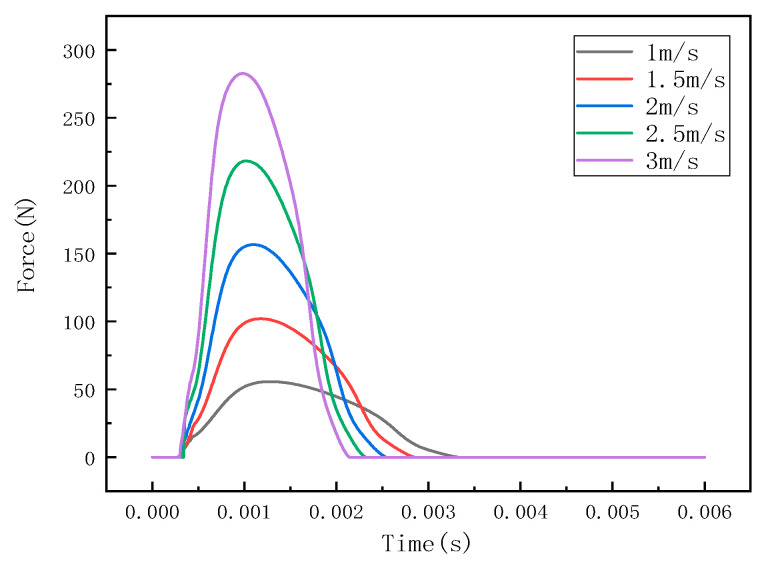
Impact force curve at position 3.

**Figure 14 sensors-22-05167-f014:**
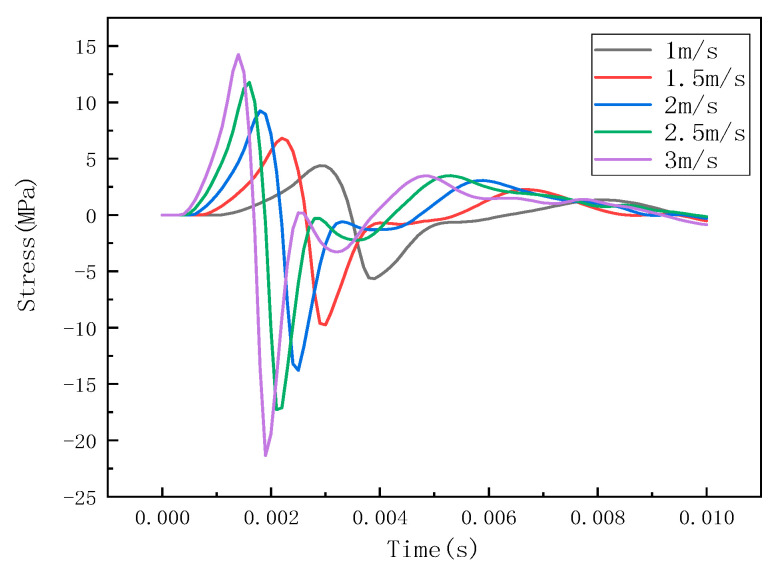
Stress curve of detection point at impact at position 3.

**Figure 15 sensors-22-05167-f015:**
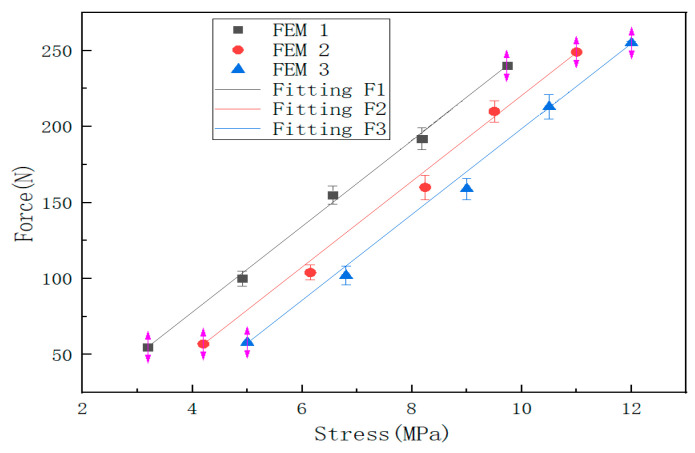
The relationship between impact force and stress response at three impact points.

**Figure 16 sensors-22-05167-f016:**
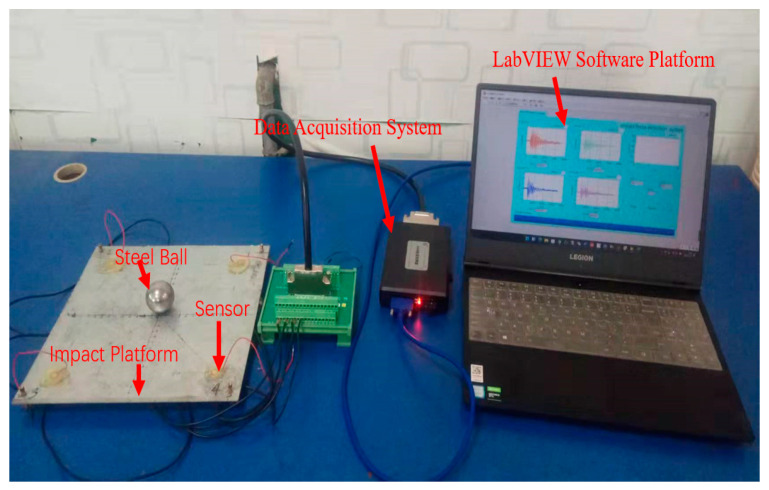
Impact detection system.

**Figure 17 sensors-22-05167-f017:**
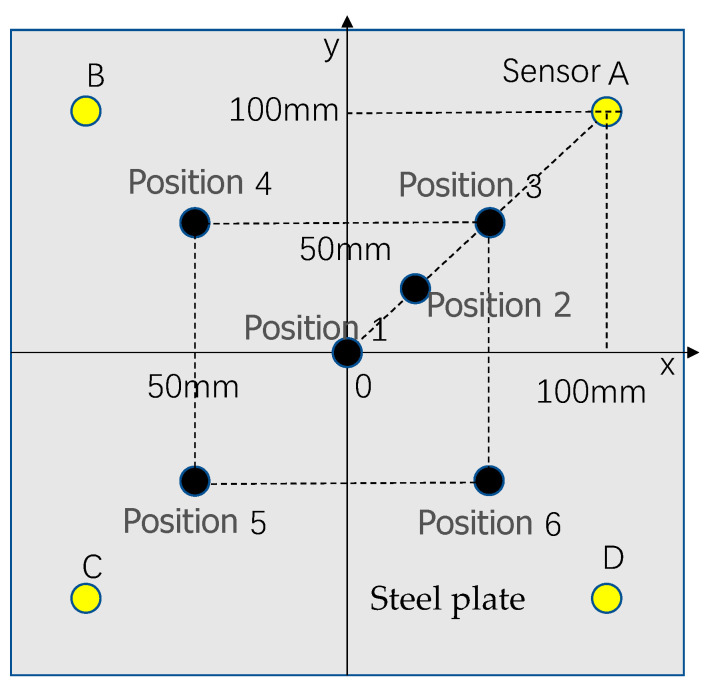
The set impact position.

**Figure 18 sensors-22-05167-f018:**
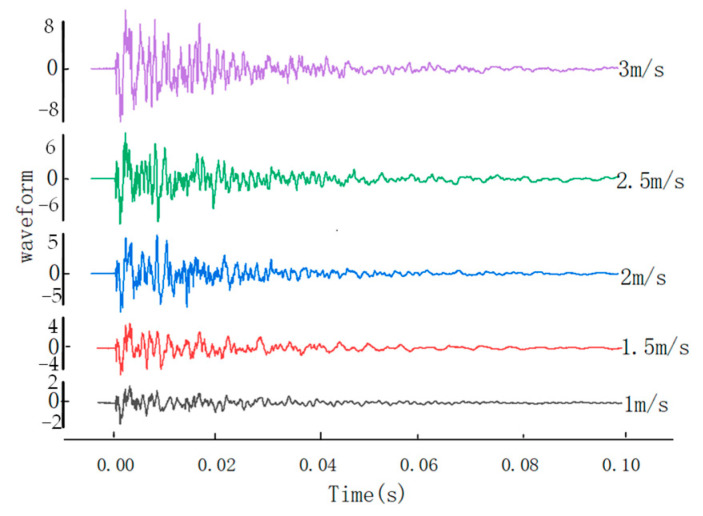
Voltage waveforms under different impact speeds.

**Figure 19 sensors-22-05167-f019:**
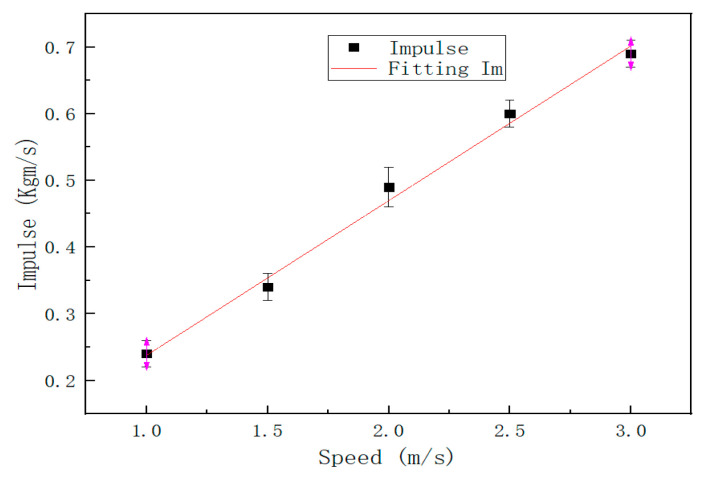
Impulse when impacting at different speeds.

**Figure 20 sensors-22-05167-f020:**
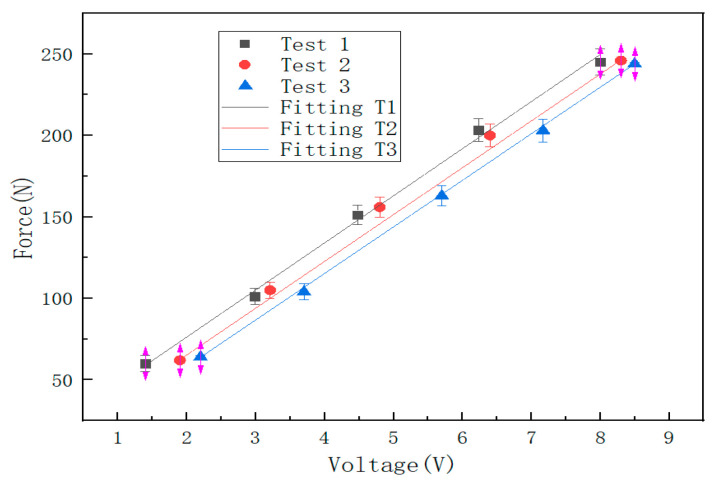
The relationship between the impact force of three impact points and the corresponding voltage signal.

**Figure 21 sensors-22-05167-f021:**
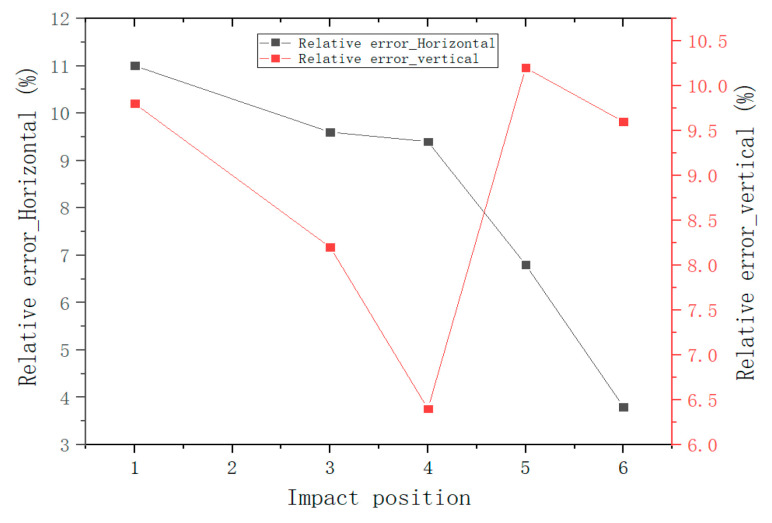
The relative errors of the impact location estimation.

**Table 1 sensors-22-05167-t001:** Finite element model and material parameters.

Model Composition Name	Parameter Category	Parameter Value
Steel ball	Diameter, *d*	40 mm
Steel plate	Side length, *v*	275 × 275 × 3 mm^3^
	Density, ρ	7850 kg/m^3^
	Young’s modulus	200 GPA
	Poisson’s ratio	0.3

**Table 2 sensors-22-05167-t002:** The functional relationship of finite element analysis shock model.

Finite Element Analysis	Fitting Equation	Goodness of Fit
Fitting F1	f1 = 28.29s1 − 35.31	*R*^2^ = 0.9999
Fitting F2	f2 = 28.25s2 − 61.95	*R*^2^ = 0.9997
Fitting F3	f3 = 28.15s3 − 82.98	*R*^2^ = 0.9998

**Table 3 sensors-22-05167-t003:** The functional relationship of the impact model of the sensor test.

Sensor Test	Fitting Equation	Goodness of Fit
Fitting T1	f1 = 28.9v1 + 18.47	*R*^2^ = 0.997
Fitting T2	f2 = 28.73v2 + 7.76	*R*^2^ = 0.9996
Fitting T3	f3 = 28.57v3 − 1.02	*R*^2^ = 0.9999

**Table 4 sensors-22-05167-t004:** Comparison of experimental results and actual positions.

Impact Position of Steel Ball	Real Coordinates (m)	Detection Coordinates (m)
Impact position 1	(0.0000, 0.0000)	(0.0055, 0.0049)
Impact position 3	(0.0500, 0.0500)	(0.0452, 0.0541)
Impact position 4	(−0.0500, 0.0500)	(−0.0547, 0.0468)
Impact position 5	(−0.0500, −0.0500)	(−0.0534, −0.0551)
Impact position 6	(0.0500, −0.0500)	(0.0481, −0.0548)

**Table 5 sensors-22-05167-t005:** Comparison of impact force obtained via numerical simulation and experimental test.

Impact Speed	FEM1	Test1	Relative Error (%)	FEM2	Test2	Relative Error (%)	FEM3	Test3	Relative Error (%)
1 m/s	55 N	60 N	9.0	57 N	62 N	8.7	58 N	64 N	10.3
1.5 m/s	100 N	101 N	1.0	104 N	105 N	0.9	102 N	104 N	1.9
2 m/s	155 N	151 N	2.5	160 N	156 N	2.5	159 N	163 N	2.5
2.5 m/s	192 N	203 N	5.7	210 N	200 N	4.7	213 N	203 N	4.6
3 m/s	240 N	245 N	2.0	249 N	246 N	1.2	255 N	244 N	4.3

## Data Availability

Not applicable.

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
