# Peer review of "A Study on Impact Force Detection Method Based on Piezoelectric Sensing"

_sensors, 2022, doi:10.3390/s22145167_

Round 1
Reviewer 1 Report
This study deals with the impact force detection method based on piezoelectric sensing. The content is suitable to publish on Sensors. The manuscript was well written. The reviewer has some comments as follows:
1) In the paragraph with lines 87-96, the novelty and contribution of the study must be clarified.
2) For the reader's convenience to follow, the theoretical basis together with the corresponding equations must be clearly presented in section 2. For example, how to determine the impact force from the stress and from the voltage?
3) In section 2.2, for the numerical simulation, please clarify the types of element used in the finite element model; and how to simulate the impact force in the FE model.
4) In line 136, the impact points (position 1, position 2, and position 3) should be positioned to the coordinates (Oxy) instead of to the distance. In Figure 3, the words of “Location” should be “Position”.
5) What does “TOA” mean in Figure 5?
6) The results for cases of Position 2 and Position 3 (similar to Figures 4 – 7) must be added and discussed.
7) In line 178, the words of “FAM1, FAM2, and FAM3” must be “FEM1, FEM2, and FEM3”.
8) What type of piezoelectric sensors were used in the experiment?
9) How to control the impact’s speeds in the experiment? Please clarify.
10) In Figure 10, the words of “Location” should be “Position”.
11) For the results in Figure 11, the number of samples and sampling frequency must be explained.
12) The difference (%) between numerical simulation and experiment, and the impact force’s unit must be added in Table 5.
Author Response
我们感谢审稿人的友好评论。下文提供了对这些意见和建议的回应。(注:这里用黑色标注了审稿人的意见,用蓝色标注了我们的回复)。详情请参阅附件。

Reviewer 2 Report
The authors proposed a method to identify impact force by piezoelectric sensing. The following issues should be addressed before it can be accepted for publication.
1. The background where the problem is coming from should be introduced.
2. The novelty or originality should be strengthened.
3. The theory of the proposed method should be intriduced in details.
4. What is the optimal number of sensors? Can three sensors successfully detect the impact?
5. Is there any requirement of boundary condition for the proposed method?
6. Is this method only applicable to the ball impact perpendicular to the plate? Can the method work for other cases?
Author Response
We thank the reviewers for their kind comments. Responses to these comments and suggestions are provided below. (Note: The reviewers' comments are marked in black here and our responses are marked in blue). Please see the attachment for details.

Round 2
Reviewer 1 Report
The manuscript has been revised following the reviewer’s comments.
Reviewer 2 Report
The reviewer has no further comments.